# Validity and Test-Retest Reliability of a Novel Push Low-Cost Hand-Held Dynamometer for Knee Strength Assessment during Different Force Ranges

**DOI:** 10.3390/diagnostics12010186

**Published:** 2022-01-13

**Authors:** Maria de Cássia Macedo, Matheus Almeida Souza, Kariny Realino Ferreira, Laura Oliveira Campos, Igor Sérgio Oliveira Souza, Michelle Almeida Barbosa, Ciro José Brito, Leonardo Intelangelo, Alexandre Carvalho Barbosa

**Affiliations:** 1Musculoskeletal Research Group—NIME, Department of Physical Therapy, Federal University of Juiz de Fora, Juiz de Fora 36036-900, Brazil; mariadecassia.macedo@hotmail.com (M.d.C.M.); malmeida_1812@hotmail.com (M.A.S.); karinyrealino@gmail.com (K.R.F.); lauracamp.1708@gmail.com (L.O.C.); igorsergio13@gmail.com (I.S.O.S.); michellecsalmeida@yahoo.com.br (M.A.B.); 2Department of Physical Education, Federal University of Juiz de Fora, Juiz de Fora 36036-900, Brazil; ciro.brito@ufjf.edu.br; 3Department of Physical Therapy, Universidad del Gran Rosario, Rosario C1021AAH, Argentina; leonardo.intelangelo@gmail.com

**Keywords:** muscle strength, knee assessment, isometric contraction

## Abstract

The objective was to assess the instrumental validity and the test–retest reliability of a low-cost hand-held push dynamometer adapted from a load-cell based hanging scale (tHHD) to collect compressive forces in different ranges of compressive forces. Three independent raters applied 50 pre-established compressions each on the tHHD centered on a force platform in three distinct ranges: ~70 N, ~160 N, ~250 N. Knee isometric strength was also assessed on 19 subjects in two sessions (48 h apart) using the tHHD anchored by an inelastic adjustable strap. Knee extension and flexion were assessed with the participant seated on a chair with the feet resting on the floor, knees, and hips flexed at 90°. The isometric force peaks were recorded and compared. The ICC and the Cronbach’s α showed excellent consistency and agreement for both instrumental validity and test–retest reliability (range: 0.89–0.99), as the correlation and determination coefficients (range: 0.80–0.99). The SEM and the MDC analysis returned adequate low values with a coefficient of variation less than 5%. The Bland–Altman results showed consistency and high levels of agreement. The tHHD is a valid method to assess the knee isometric strength, showing portability, cost-effectiveness, and user-friendly interface to provide an effective form to assess the knee isometric strength.

## 1. Introduction

Muscle weakness increases the risk of injuries on different populations [1,2,3]. The occurrence of injuries due to muscle weakness impairs functional independence, sports practice, leading to increased costs to public health system [4,5,6,7]. As part of physical assessment, the maximal isometric strength is used as an objective parameter to prescribe exercise and to evolve the exercise training [8,9,10,11]. Additionally, several studies reported the isometric strength ability to predict the occurrence of non-contact injuries or even the higher incidence of joint pain [12,13,14,15]. To perform those objective assessments, the clinician or the coach must use a device that provides the force output in kilogram or in Newton. However, the gold-standard equipment (i.e., the isokinetic dynamometer) [1,2] is expensive, not portable, requires extensive staff training, and it is limited to laboratory environment.

Inexpensive, accurate and more affordable devices are then essential to objectively assess isometric muscle strength [16]. Thus, hanging scales, load-cell transducers and hand-held dynamometers (HHD) have been proposed as valid and reliable alternatives [7,17,18,19,20]. Despite their usefulness, portability and the efforts to ensure the relative accurate measures obtained from those devices, most previous studies only described the overall tension or compression force outputs applied on the equipment, not distinguishing the acquired precision during different ranges of load [7,17,18,19]. The lack of precision from those measurements may lead to misinterpretation during softer compared to heavier loads applied on the device (i.e., accuracy to differentiate weak vs. strong/normal muscles). Other concerns were raised during a test–retest study using the isokinetic, a fixed load-cell type dynamometer, and a portable HHD [20]. Test–retest reliability assessed between days for knee extension was considered high for the first and second devices while fair reliability was demonstrated using the HHD. Despite the price of those devices is a fraction of a gold-standard isokinetic, they are still expensive (from USD 1000 to USD 5000) for most clinicians.

Due to the above-mentioned issues, the objective strength assessment is essential to establish prospective evaluation, compare baseline results to other timeline assessments, or even as a prognostic measure to predict future outcomes [21,22]. Thus, the present study aimed to assess the instrumental validity and the test–retest reliability of a low-cost push hand-held dynamometer (~USD 160) adapted from a load-cell based hanging scale to collect compressive forces, emulating a commercially available HHD in different ranges of compressive forces.

## 2. Materials and Methods

### 2.1. Equipment

All data were collected at the facilities of the Clinic-School of Physical Therapy, Federal University of Juiz de Fora. The tested dynamometer—tHHD (MED.DOR Ltd., Governador Valadares, Brazil; maximum compression = 2000 N, 4-digit display (Figure 1)) calibration was checked by placing 5 known weights (50–250 N) on the application surface. The maximal tolerated difference between the weight and the value on the display was 1 N. The tHHD used in the present study was brand new, and the calibration was checked twice before any measurement. 

A gold-standard two-axis force platform (37 cm × 37 cm; Pasport PS-2142; PASCO, Roseville, CA, USA) collected data using five force beams (sample rate = 1000 Hz). Four beams in the corner were used to measure the vertical force (range: −1100 N to +4400 N) and a 5th beam measured the force in a parallel axis (range: −1100 N to +1100 N).

### 2.2. Procedures

#### 2.2.1. Instrumental Validity

Three independent raters performed fifty 3-s pressure trials each using the tHHD centered on the force platform (Figure 2). An off-board USB camera was synchronized and positioned facing the dynamometer’s display to record the peak values. Each rater manually applied progressive pressures on the tHHD until reaching a previously determined threshold (1st rater’s threshold ~70 N; 2nd rater threshold ~160 N; and 3rd rater threshold ~250 N). The raters were allowed to guide their pressure application using the tHHD display. The threshold pressure was kept for 3 s. All 3 raters were blinded to the force platform’s results. Data were collected and stored using the PASCO Capstone Software (Version 1.13.4; PASCO Scientific, 2019, Roseville, CA, USA). The maximal peaks were then extracted offline from the force platform software recordings and from the tHHD recordings.

#### 2.2.2. Test–Retest Reliability

A convenience sample of 25 participants (24.21 ± 4.06 years; 1.70 ± 0.07 m; 67.83 ± 14.03 kg) were recruited by the public invitation through folders and personal contacts. The a priori two-tailed point biserial model sample size calculation was performed using the G-power 3.1 Software (Franz Faul, Univesity Kiel, Germany) considering a coefficient of determination of 0.97 with an effect size of 1.04 obtained from a previous similar study [18], with an alpha of 5% and a sampling power (1-β) of 95%. A sample size of 15 subjects was returned with an actual power of 0.962. Exclusion criteria included a history of injury on the lower extremity during the past six months, a history of hip and knee osteoarthritis, previous knee surgery, diagnosed neurologic disorder (e.g., stroke, head trauma), or current symptoms related to the hip and knee area. The UFJF ethics committee for human investigation approved (Number of Approval: 29238720.7.0000.5147) the procedures employed in the study. The objectives, benefits and potential risks involved were previously explained to all participants. Then, they all signed an informed consent form before participation.

After an initial familiarization session and following a warm-up set of submaximal bilateral isometric knee’s flexion-extension, the participants were asked to perform two sessions (Day 1 and Day 2) of 3 trials of maximal flexion-extension isometric contractions (3 min of rest between trials; 48 h between sessions). Each subject was asked to refrain from strenuous exercise or training 48 h before assessments and to avoid eating 2 h before testing. During the test, the participants remained seated on a chair with their arm crossed on the chest. The knee flexed at 90° with the feet resting on the floor, with hips flexed at 90°. All angles were quantified through goniometric measurements. An adjustable inelastic strap was then anchored on a metallic bar as the dominant lower limb was involved by the same strap. The tHHD was positioned between the strap and the posterior distal portion of the leg (right above the malleolus line) for flexion and anteriorly for extension. The volunteer was instructed to perform three maximum isometric contractions trying to flex-extend the knee. Verbal encouragement was given to ensure maximal effort (push, keep pushing, stop). The peaks from each trial were extracted and the means were used for statistical analysis.

### 2.3. Statistical Analysis

Data were presented as mean and standard deviation. The Shapiro–Wilk’s and the Levene’s tests were used to test the data distribution and the homoscedasticity, respectively. The normality and the homogeneity were both accepted. Significance was set at *p* < 0.05. Two-way mixed effects model intraclass correlation coefficient (ICC) was calculated to assess the reliability between results [23]. The Cronbach’s α test was used to assess the expected correlation measuring the same construct. ICC and Cronbach’s α values were qualitatively classified as poor (<0.50), moderate (0.5–0.75), good (0.75–0.90) or excellent (>0.90) [23]. Linear regression estimated the coefficient of correlation (r) and the adjusted coefficient of determination (r^2^). The correlation coefficients were qualitatively classified as high (>0.70), moderate (0.50–0.70), low (0.30–0.50) and weak (<0.30) [24]. The Bland–Altman method estimated the measurement bias, with lower and upper limits of agreement between results. Standard error of measurement (SEM), percentage of SEM as a coefficient of variation (%SEM = SEM × 100/mean of Day 1 and Day 2), and minimal detectable change at a 95% confidence level (MDC = SEM × 1.96 × √2) were calculated. A %SEM of 10% or less was set as the level at which a measure was considered reliable [25,26]. All statistics were performed using the JAMOVI software. (The JAMOVI project [2021]. Version 1.6. Retrieved from https://www.jamovi.org (accessed on 11 March 2021)).

## 3. Results

### 3.1. Validity Analysis

Descriptive and validity data for all force variables are presented in Table 1. The ICC and the Cronbach’s α showed excellent consistency and agreement (>0.95). The results also showed excellent correlation and determination coefficients between the force platform and the tHDD (>0.97). The SEM ranged from 0.14 to 1.20, with %SEM less than 2%, suggesting the tHHD as a reliable measure compared to the force platform. The MDC analysis returned a range from 0.38 to 3.32 N. The Bland–Altman results showed high levels of agreement (Figure 3). 

### 3.2. Reliability Analysis

Descriptive and test–retest reliability is presented in Table 2. ICC and Cronbach’s α showed good to excellent results (range: 0.89–0.97), with high levels of between-day correlation. The SEM and the MDC analysis returned adequate low values with a coefficient of variation less than 5%. The Bland–Altman results showed consistency and high levels of agreement (Figure 4).

## 4. Discussion

The present findings showed the validity of tHHD, not only considering the overall applied force output, but also the different ranges to differentiate weak from strong muscles during knee isometric flexion and extension. This is particularly important considering the price range of the commercial HHD compared to tHHD, and the fact that, to our knowledge, this is the first study to consider those distinct ranges. Strength improvements should be measured along training to fulfill the subject’s needs and to assure adequate training adjustments [18]. Isometric strength re-assessments are valid and effective to evaluate the strength-training adaptations, as the torque production measured at a constant angle is more sensitive to influences produced by muscle fiber type rather than angle-independent peak torque during dynamic contractions [16]. For rehabilitation routines the validity in distinct ranges of force constitutes an essential component to consider during prospective assessments, as musculoskeletal injuries/diseases often provoke muscle weakness and deficits on the force output, while training promotes the recovery of strength and consequent changes on the isometric maximal force. For the clinician, the precision of each assessment is crucial to decide whether to evolve (or not) the load levels along with the training session. Additional to the validity, the present test–retest reliability showed optimal results to prospectively assess the isometric knee strength. 

The current results are consistent with previous studies that assessed hanging scales as cost-effective alternatives to ensure accuracy and safe performance for muscle strength assessments during isometric knee movements [7,18]. Those studies showed excellent ICC (>0.90) in every assessed movement comparing the tested device to isokinetic dynamometer or laboratorial load-cells. Intra and inter-tester reliability was also good for all the movements assessed (ICC > 0.75). Those hanging scales were also validated to assess other joint isometric strength, such as shoulder, elbow, hip and ankle, during distinct ranges of force loads [7,17,18,19]. To collect force data, the hanging scales have consistently shown accuracy and reliability. However, all studies considered only the devices’ tension function during movements exerted in traction, without any adaptation to handle the device. This may impair the usefulness in daily routine due to time to set-up the device and adjust the anchoring while positioning the patient. The present tHHD terminals were adapted to be used with a common adjustable inelastic strap, minimizing the time to anchor and to set the device up. The tHHD might also be useful to detect asymmetries during the assessments, between limbs and also monitor the muscle strength while the training protocol evolves, collaborating to decrease the risk of injuries and reduce the costs associated to musculoskeletal weaknesses [1,27]. Additionally, and based on another study [4], the authors emphasize the importance of limb positioning to perform strength tests, as changes on positioning the joint might influence the joint’s ability to develop muscle force. The present results showed excellent validity and reliability for knee movements on seated position. Thus, we recommend the maintenance of the described participant’s body positioning, and the tHHD anchoring to ensure the same results. In the present study, the novelty is the adaptation of a hanging scale to provide a push type dynamometer. Other studies only used the hanging scale in a pull mode [3,4,5,6]. In addition, we assessed the precision of the device in distinct ranges of compression. All results showed the validity for such analysis, not presented in previous studies. Finally, our study provides a cheaper device to assess muscle strength. This is especially important in developing and poor countries, where the cost of a commercial dynamometer is prohibitive.

The tHHD validity to assess knee isometric strength would allow health professionals and coaches to objectively evaluate strength with less complexity, as no training is required to use the device. Minimal investment is also an advantage, as the device is not expensive compared to other equipment, such as isokinetic, and laboratorial load-cells. However, some limitations must be acknowledged. The present study included only healthy and young participants. The results may differ in the presence of pathology. Nevertheless, to avoid any adverse outcome due to repeated trials, the safety of the procedure should be firstly assessed with non-pathological subjects, as we did. Another limitation is that the tHHD provides only the peak force, while other devices would allow the extraction of other measures such as rate of force development and time to peak force, that cannot be measured using the tHHD. The participants’ movements were assessed in specific positions. Of note, the positioning is an essential factor that may affect the joint’s ability to produce force. The present results may vary accordingly. It was not our goal to establish the validity for all joint movements. Instead, to provoke other studies that assess distinct joint movements in other population, as many other studies previously did [3,5]. 

## 5. Conclusions

The results suggest the tHHD as a valid and reliable method to assess the knee isometric strength. The portability, the cost-effectiveness, and the user-friendly interface provide an effective form to assess the knee isometric strength.

## Figures and Tables

**Figure 1 diagnostics-12-00186-f001:**
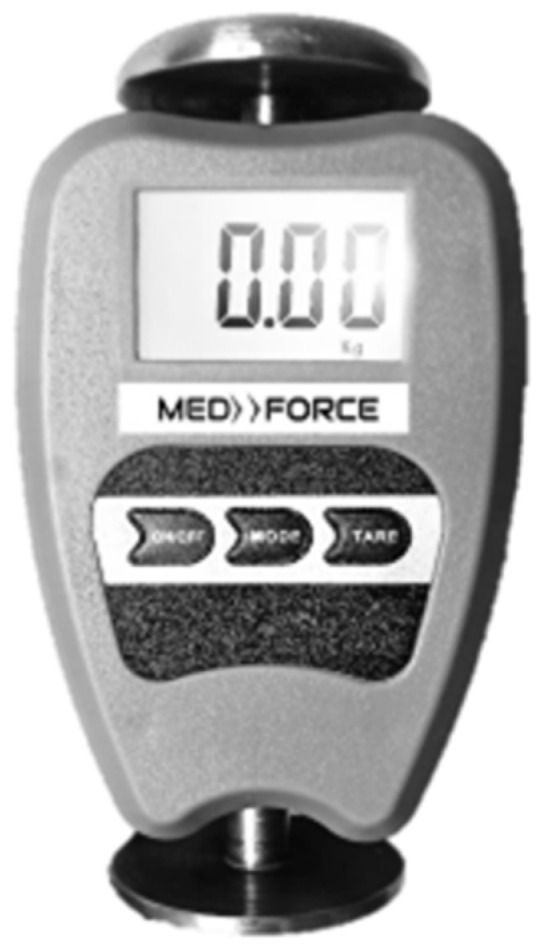
The push low-cost hand-held dynamometer.

**Figure 2 diagnostics-12-00186-f002:**
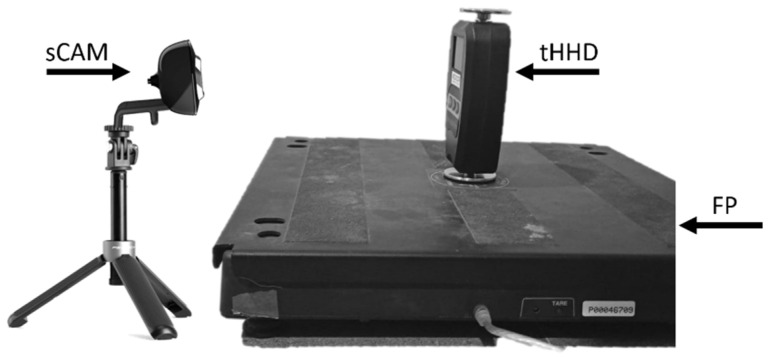
Experimental set-up. tHHD = tested dynamometer; FP = force platform; sCAM = synchronized camera.

**Figure 3 diagnostics-12-00186-f003:**
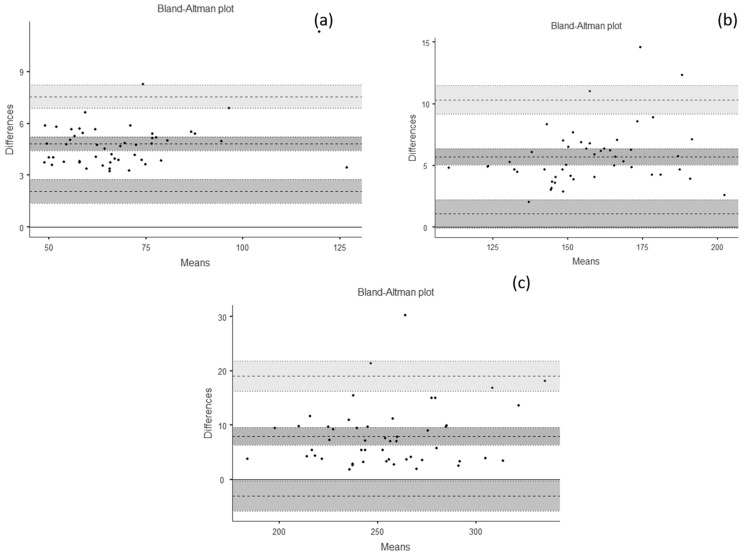
Validity analysis—Bland–Altman Plots. (**a**) ~70 N: bias = 4.81 (95% confidence interval [CI] = 4.41 to 5.21); lower limit of agreement (LLA) = 2.06 (95% CI = 1.37 to 2.74); upper limit of agreement (ULA) = 7.56 (95% CI = 6.88 to 8.25). (**b**) ~160 N: Bias = 5.70 (95% CI = 5.03 to 6.38); LLA = 1.07 (95% CI = −0.08 to 2.22); ULA = 10.34 (95% CI = 9.18 to 11.50). (**c**) ~250 N: bias = 7.99 (95% CI: 6.38 to 9.58); LLA = −3.06 (95% CI: −5.81 to −0.30); ULA = 19.03 (95% CI: 16.27 to 21.78).

**Figure 4 diagnostics-12-00186-f004:**
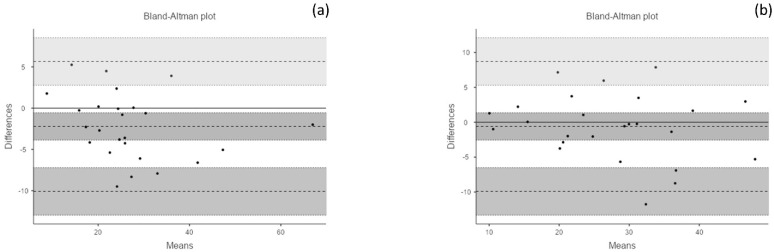
Test–retest reliability analysis—Bland–Altman Plots. (**a**) Flexion: bias = −2.21 (95% confidence interval [CI] = −3.87 to −0.55); lower limit of agreement (LLA) = −10.08 (95% CI = −12.95 to −7.21); upper limit of agreement (ULA) = 5.65 (95% CI = 2.78 to 8.52). (**b**) Extension: bias = −0.60 (95% CI = −2.56 to 1.36); LLA = −9.91 (95% CI = −13.31 to −6.52); ULA = 8.70 (95% CI = 5.31 to 12.10).

**Table 1 diagnostics-12-00186-t001:** Validity analysis.

Outcome	tHHD (in N)	Force Platform (in N)	ICC	Cronbach α	r	r^2^	SEM	%SEM	MDC (in N)
Overall	157.03 ± 79.19	163.19 ± 80.67	0.999	0.999	0.999	0.998	0.14	0.09	0.38
~70 N	65.92 ± 15.97	70.73 ± 16.48	0.954	0.998	0.997	0.993	0.73	1.07	2.02
~160 N	154.36 ± 19.29	160.06 ± 19.93	0.953	0.996	0.993	0.987	0.87	0.56	2.42
~250 N	250.80 ± 31.48	258.79 ± 32.56	0.955	0.992	0.985	0.970	1.20	0.47	3.32

Legend: tHHD = push hand-held dynamometer; ICC = intraclass correlation coefficient; r = coefficient of correlation; r^2^ = coefficient of determination; SEM = standard error of measurement; MDC = minimal detectable change.

**Table 2 diagnostics-12-00186-t002:** Reliability analysis.

Outcome	Day 1 (in N)	Day 2 (in N)	ICC	Cronbach α	r	r^2^	SEM	%SEM	MDC (in N)
Flexion	253.03 ± 112.03	274.75 ± 122.14	0.930	0.971	0.947	0.897	0.90	3.33	2.48
Extension	266.84 ± 96.39	272.76 ± 106.22	0.897	0.944	0.899	0.808	1.00	3.65	2.78

Legend: ICC = intraclass correlation coefficient; r = coefficient of correlation; r^2^ = coefficient of determination; SEM = standard error of measurement; MDC = minimal detectable change.

## Data Availability

The raw data presented in this study are openly available in doi: 10.17632/8bj9kbkf9b.1.

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
