# Peer review of "Validity and Test-Retest Reliability of a Novel Push Low-Cost Hand-Held Dynamometer for Knee Strength Assessment during Different Force Ranges"

_diagnostics, 2022, doi:10.3390/diagnostics12010186_

Round 1
Reviewer 1 Report
Thank you for the opportunity to review this manuscript entitled “Validity and test-retest reliability of a novel push low-cost hand-held dynamometer for knee strength assessment during different force ranges”. This is one research for studying the validity and precision in different range of load in the knee measured by hand-held dynamometer.
The authors have conducted a laboratory study to assess the instrumental validity and the test-retest reliability.
Overall, I found some aspects should be reviewed.
I think this is good research to improve our clinical assessment possibilities of strength in clinical environment.
I propose some minor corrections.
Line 82-91: explain in more detail this procedure.
Author Response
RESPONSE TO REVIEWER 1:
Thank you for the opportunity to review this manuscript entitled “Validity and test-retest reliability of a novel push low-cost hand-held dynamometer for knee strength assessment during different force ranges”. This is one research for studying the validity and precision in different range of load in the knee measured by hand-held dynamometer.
The authors have conducted a laboratory study to assess the instrumental validity and the test-retest reliability.
Overall, I found some aspects should be reviewed.
I think this is good research to improve our clinical assessment possibilities of strength in clinical environment.
I propose some minor corrections.
Line 82-91: explain in more detail this procedure.
ANSWER: Thank you for your comment. We restructured and added more information about the procedure. A figure was also added. Please, see our revised version.

Reviewer 2 Report
Two aspects must be resolved before a more in-depth review. First, following that described by Streiner and Norman, reliability studies based on confidence intervals (CIs), with the number of instruments (k) equal to 2, the CI around r (the reliability coefficient) of 0.05, and an estimated r of 0.95, the sample size (n) required is minimum of 25 participants. Therefore, the authors must justify their sample size in a better way, or else, expand the sample of their results. If not, these cannot be considered. Second, the authors propose a low cost dynamometer to measure knee force. Previous studies have already done this, in addition, in a greater study by incorporating more movements and population. Therefore, as long as the authors do not justify what the study of a new HHD contributes, the clinical applications are irrelevant.
Rodrigo Martín-San Agustín , Josep C. Benítez-Martínez , Lorenzo
Castillo-Ballesta , Mariano Gacto-Sánchez & Francesc Medina-Mirapeix (2020) Validity,
Reliability, and Sensitivity to Change of DiCI for the Strength Measurement of Knee and Hip
Muscles, Measurement in Physical Education and Exercise Science, 24:4, 303-311, DOI:
10.1080/1091367X.2020.1822363
Romero-Franco, N., Jiménez-Reyes, P., & MontañoMunuera, J. A. (2017). Validity and reliability of a low-cost
digital dynamometer for measuring isometric strength of
lower limb. Journal of Sports Sciences, 35(22), 2179–2184.
https://doi.org/10.1080/02640414.2016.1260152
Author Response
RESPONSE TO REVIEWER 2:
Two aspects must be resolved before a more in-depth review. First, following that described by Streiner and Norman, reliability studies based on confidence intervals (CIs), with the number of instruments (k) equal to 2, the CI around r (the reliability coefficient) of 0.05, and an estimated r of 0.95, the sample size (n) required is minimum of 25 participants. Therefore, the authors must justify their sample size in a better way, or else, expand the sample of their results. If not, these cannot be considered.
ANSWER: Thank you for your comment. As you could notice in our 1st version, we performed a sample size calculation as follows: “The a priori two-tailed point biserial model sample size calculation was performed using the G-power 3.1 Software (Franz Faul, Univesity Kiel, Germany) considering a coefficient of determination of 0.97 with an effect size of 1.04 obtained from a previous similar study [18], with an alpha of 5% and a sampling power (1-β) of 95%. A sample size of 15 subjects was returned with an actual power of 0.962.” Also, other studies with similar sample sizes were already done (https://pubmed.ncbi.nlm.nih.gov/33023868/; https://pubmed.ncbi.nlm.nih.gov/19558384/; https://pubmed.ncbi.nlm.nih.gov/31191956/) with similar sample sizes. Nevertheless, we included 6 more subjects in our analysis to fulfill the 25 minimum suggested. As a consequence, the results improved as described in our results section.
Second, the authors propose a low cost dynamometer to measure knee force. Previous studies have already done this, in addition, in a greater study by incorporating more movements and population. Therefore, as long as the authors do not justify what the study of a new HHD contributes, the clinical applications are irrelevant.
ANSWSER: Thank you again for your comment. However, we must humbly disagree. There are several studies that continuously assess new equipment to ensure their validity and reliability in distinct ways. We believe that this is necessary, as technology becomes more affordable and provide the chance to adapt devices to novel applications in clinical settings. In the present case, the novelty is the adaptation of a hanging scale to provide a push type dynamometer. Other studies, including those cited by the reviewer, only used the hanging scale in a pull mode. Also, we assessed the precision of the device in distinct ranges of compression. All results showed the validity for such analysis, not presented in previous studies. Finally, our study provides a cheaper device to assess muscle strength. This is especially important in developing and poor countries, where the cost of a commercial dynamometer is prohibitive due to currency exchange. It is not our goal to establish the validity for all movements, as stated in our title and along with the text. But instead to provoke other studies that assess distinct joint movements in other population, as many other studies previously did (https://pubmed.ncbi.nlm.nih.gov/26519103/; https://pubmed.ncbi.nlm.nih.gov/26509265/; https://pubmed.ncbi.nlm.nih.gov/29596450/; https://pubmed.ncbi.nlm.nih.gov/19558384/).
Reviewer 3 Report
Comments to the Author
SUMMARY:
The aim of the article: “Validity and test-retest reliability of a novel push low-cost hand-2 held dynamometer for knee strength assessment during different force ranges” is to assess the instrumental validity and the test-retest reliability of a 14 low-cost hand-held push dynamometer.
GENERAL COMMENTS:
This is a well-written manuscript. Study design and Statistical analyses are appropriate. Results are also clearly presented.
Comments :
Line 39 : What are the references regarding the gold-standard equipment?
Line 93: Can you specify how objective results can be obtained from a group of 19 subjects
I would recommend being more precise at the Abstract section by adding the numerical results and by expanding the Discussion section
Author Response
RESPONSE TO REVIEWER 3
The aim of the article: “Validity and test-retest reliability of a novel push low-cost hand-2 held dynamometer for knee strength assessment during different force ranges” is to assess the instrumental validity and the test-retest reliability of a 14 low-cost hand-held push dynamometer.
GENERAL COMMENTS:
This is a well-written manuscript. Study design and Statistical analyses are appropriate. Results are also clearly presented.
ANSWER: Thank you for your kind comment.
Comments :
Line 39 : What are the references regarding the gold-standard equipment?
ANSWER: Thank you for your comment. We added references.
Line 93: Can you specify how objective results can be obtained from a group of 19 subjects
ANSWER: Thank you for your comment. We performed a sample size calculation, as described in our previous version, and followed previous similar studies that used this amount of participants ((https://pubmed.ncbi.nlm.nih.gov/33023868/; https://pubmed.ncbi.nlm.nih.gov/19558384/; https://pubmed.ncbi.nlm.nih.gov/31191956/). However, in response to another reviewer, we added 6 more subjects to our sample to ensure a minimum of 25 participants, as requested.
I would recommend being more precise at the Abstract section by adding the numerical results and by expanding the Discussion section
ANSWER: Thank you for your comment. Due to space constraints, we added the range values of ICC/Cronbach’s alfa, coefficient of correlation/determination. The discussion section was expanded as suggested. Please, see our new version.

Round 2
Reviewer 2 Report
Ok